# Understanding the impact and causes of 'failure to attend' on continuity of care for patients with chronic conditions

Amy-Louise Byrne[1]*, Adele Baldwin[1], Clare Harvey[1], Janie Brown[1,2,3], Eileen Willis[1,4], Desley Hegney[5,6], Bridget Ferguson[6], Jenni Judd[7], Doug Kynaston[8], Rachel Forrest[9], Brody Heritage[10], David Heard[1], Sandy Mclellan[11], Shona Thompson[12], Janine Palmer[13]

1 School of Nursing Midwifery and Social Science, Central Queensland University, Townsville, Queensland, Australia, 2 School of Nursing, Midwifery and Paramedicine, Curtin University, Perth, Australia, 3 St John of God Midland Public and Private Hospital, Midland, Australia, 4 College of Nursing and Health Science, Flinders University, Adelaide, Australia, 5 School of Nursing, The University of Adelaide, Adelaide, Australia, 6 School of Nursing, Midwifery and Social Sciences, Central Queensland University, North Rockhampton, Australia, 7 School of Health Medical and Applied Sciences, Central Queensland University, Bundaberg, Australia, 8 Sunshine Coast Hospital and Health Service, Birtinya, Australia, 9 School of Nursing, Eastern Institute of Technology, Napier, New Zealand, 10 Discipline of Psychology, College of Science, Health, Engineering and Education, Murdoch University, Perth, Australia, 11 School of Nursing Midwifery and Social Science, Central Queensland University, Mackay, Queensland, Australia, 12 Eastern Institute of Technology, Napier, New Zealand, 13 Hawke's Bay District Health Board, Hastings, New Zealand

* a.byrne@cqu.edu.au

**Data Availability Statement:** All relevant data are within the manuscript and its Supporting information files.

## Abstract

### Aim

To understand the impact and causes of 'Failure to Attend' (FTA) labelling, of patients with chronic conditions.

### Background

Nurse navigators are registered nurses employed by public hospitals in Queensland, Australia, to coordinate the care of patients with multiple chronic conditions, who frequently miss hospital appointments. The role of the nurse navigator is to improve care management of these patients. Evidence for this is measured through improvement in patient self-management of their conditions, a reduction in preventable hospital admissions and compliance with attendance at outpatient clinics. Failure to attend (FTA) is one measure of hospital utilisation, identifying outpatient appointments that are cancelled or not attended.

### Method

The cohort for this study was patients with multiple chronic conditions, and nurse navigators coordinating their care. Data describing the concept of FTA were thematically analysed twelve months into this three year evaluation.

### Results

Although the patient is blamed for failing to attend appointments, the reasons appear to be a mixture of systems error/miscommunication between the patient and the health services or

**Funding:** Funding (financial support) for this project was provided by Queensland Health. The funders had no role in study design, data collection and analysis, decision to publish, or preparation of the manuscript. No other funding was received for the project. No authors have received a salary from the funders.

**Competing interests:** The authors have declared that no competing interests exist.

social reasons impacting on patient's capacity to attend. Themes emerging from the data were: access barriers; failure to recognise personal stigma of FTA; and bridging the gap.

## Conclusion

The nurse navigators demonstrate their pivotal role in engaging with outpatient services to reduce FTAs whilst helping patients to become confident in dealing with multiple appointments. There are many reasons why a patient is unable to attend a scheduled appointment. The phrase 'Failure to Attend' has distinctly negative connotations and can lead to a sense of blame and shame for those with complex chronic needs. We propose the use of the neutral phrase "appointment did not proceed" to replace FTA.

### Implications for Nursing management

This article advocates for further consideration of collaborative models that engage the patient in their care journey and for consideration of the language used within the outpatient acute hospital setting, proposing the term 'appointment did not proceed.'

## Introduction

Labelling patients who miss an appointment with a healthcare provider as a "failure to attend" (FTA) may have unintended negative consequences on their future healthcare engagement and subsequent quality of life. The FTA phenomenon is not unique to the context within which this paper sits; it is well documented in the existing literature, most commonly in discussions about finite healthcare resources, inefficiencies and waste [1]. Further, it is not unique to healthcare. FTAs also appear in the criminal justice system, whereby people who do not appear as scheduled in court are labelled FTA. The difference in that context is that failure to attend may well result in incarceration [2]. FTA in the healthcare setting does not incur this type of penalty but has been reported to make the patient feel like they are in purgatory [3]. The current evidence shows that health providers have seriously considered other punitive measures for FTA patients, such as imposing a financial fine, or implementing reminder strategies such as SMS reminders about impending appointments [1, 3]. Consistent across the this research is the frequency of patients 'forgetting' appointments and the relationship between referral time to actual appointment, with limited research into the patient characteristics that may predispose them to failing to attend.

Patients living in the community with chronic, complex co-morbidities is recognised as the preferred model of care and is being adopted across the globe in an effort to make healthcare provision more cost effective and therefore sustainable [4]. However, the link between disengagement with healthcare in the form of FTAs for people with chronic, complex diseases is not apparent. People living with chronic, complex diseases are particularly vulnerable to flaws in the system and are arguably some of the most vulnerable people in the community [4]. It is therefore important that the relationship between living with chronic, complex disease and failing to attend outpatient appointments is explored further. Better understanding is required about the reasons for FTA, and the impact of being labelled FTA, so that appropriate interventions to address organisational processes can be recommended. This paper reports specifically on the phenomenon of FTAs as part of a larger study evaluating the role of the nurse navigators.

## Background

People living with complex chronic conditions require care and intervention across multiple levels of health service and treatment modalities, from specialist clinical services through to community care services [5]. Internationally, nurse-led coordination is becoming a popular approach to care for patients with multiple conditions, in an effort to reduce costs, support continuity in care and to provide a more individualised service, based on the long-term needs of such patients [6, 7]. In Queensland, Australia, nurse navigators provide the continuity of care for people living with chronic conditions, helping them to navigate the system and manage their condition, within a framework of person-centred care [8].

As per the nurse navigator suggested patient enrolment criteria, a person with ten or more outpatient appointments in the past year may warrant navigation [8] (p. 25). There are 16 principles outlined in the Queensland Health service standard for the management of outpatient appointments [9]. The aim is to "have patients and their carers as the primary focus", ensuring that treatment is "as close to home as possible" and "provided by the most appropriate clinician for the level required" (p. 3). The principles use words such as "empower[ring] the patient to participate in the decision making and to make informed choices on their pathway of care", and that the service is a "shared responsibility" between the health service, specialists and referring medical practitioner (p. 3).

## Aims and objectives

This paper presents data collected from nurse navigators and their patients, as part of the overall evaluation study. The impetus to explore the FTA phenomenon qualitatively was based on the interim quantitative data analysis. Subsequently, during initial qualitative data analysis the term FTAs became apparent. The emergence of FTA as a phenomenon in both data sets, prompted a reanalysis of the qualitative data to focus specifically on FTAs. The aim of the reanalysis was to understand the impact and causes of 'Failure to Attend' (FTA) labelling of patients with chronic conditions.

The research questions were:

1. How do the nurse navigators and their patients use the term FTA?

2. What are the factors that lead to a patients' failure to attend?

3. What is the reported impact of being labelled on the patients' wellbeing?

## Research design and methods

This nurse navigator service evaluation is underpinned by a Theory of Change (ToC) framework, which allows researchers to collect data, review, evaluate and refine information at regular intervals throughout the project [10]. The framework is a "process and a product", allowing the participation of the stakeholders throughout the life of the project [11]. ToC can be used where political, social and economic factors may impact on the project outcomes and where the findings need to examine long term sustainability [12]. It is also applied when a project is complex, or data collection is not linear [11]. Data collection usually includes mixed methods where multiple sources of information can be drawn upon at various stages throughout the life of the project. This allows for flexibility in what is collected, making it dynamic, responsive and iterative [13]. In our case, the flexible ToC approach allowed the research team to return to the data to further explore the concept of FTAs.

All 16 hospital and health services across Queensland Health consented to participate in the evaluation of the nurse navigator service. For the two-year nurse navigator evaluation, metrics include hospital data (emergency department and in-patient admissions, length of hospital stay, discharge against medical advice and FTA events); and surveying patient self-reported wellbeing and quality of life indicators at regular intervals [14, 15]. The evaluation also examines the effectiveness of the nurse navigator service through policy and practice development, case studies, vignettes and narratives collected through group and individual interviews (face to face and via teleconferencing) with consented nurse navigators and patients. To date, 25 individual and 14 group interviews have been completed with NN. These interviews included NNs across all areas of Queensland and are inclusive of special interest groups including a rural and remote group, midwifery group, aged care group and nurse navigators/practitioners' group. Additionally, 49 interviews with consented NN patients were conducted across all areas of Queensland. This includes patients across all life stages (from paediatrics to aged care), all of whom suffer from chronic complex conditions and frailty and are under the care of a NN. Interviews are recorded and transcribed by a professional transcription service. This deidentified and collated data were used for this research.

Using a thematic analysis underpinned by Saldana [16] and Waring and Wainwright [17], the research team reviewed all the qualitative data for any reference to FTA by patients or nurse navigators. While questions around FTA had not been specifically asked in interviews, the topic came up in several interviews, particularly with the nurse navigators because it was central to the their work and they saw it as part of the explanation for patient's alienation from the system. Initial coding was independently completed by two members of the research team (author one is a PhD candidate and author three holds a PhD) who coded the data by single word, paragraph and full content [14]. Second cycle coding was then completed by seven research team members collaboratively, (five of whom hold a PhD, the remaining two being PHD candidates) and codes were reconfigured, recategorized, resynthesized and further developed within the analytic lens of FTA [16]. Final consensus of themes was achieved collaboratively within the research team with collated themes collapsed into major themes with accompanying quotes to demonstrate the themes (See S1 Checklist). The majority of quotes come from the nurse navigators, but where possible we have added patient comments as a form of triangulation.

Analysis of the themes also drew on two Queensland Health documents; the nurse and midwife navigator Toolkit [8], and the Specialist Outpatient Implementation Standard [9]. These were used to compare and contrast the findings from patient and nurse navigator narratives, in relation to the standard of expected practice of the nurse navigator and in the management of outpatient appointments, with particular reference to FTA events.

A multiple site ethical approval was awarded to the project in March 2018 by Darling Down Research Ethics Committee HREC/18/QTDD/8.

## Findings

It was apparent from the quantitative data that the counts of FTA did not vary based on time-related change around the patients' ($n$ = 52) navigation intervals (Fig 1). Site-related variability appeared to account for the approximately 17% of the variability in FTA counts, suggesting that navigation did not meaningfully reduce these FTA counts over time. This was the only measure where there was no clear evidence of how nurse navigation contributes to a reduction in preventable hospital presentations. Fig 1 outlines the count of FTA over time both before and after engaging with the nurse navigator service.

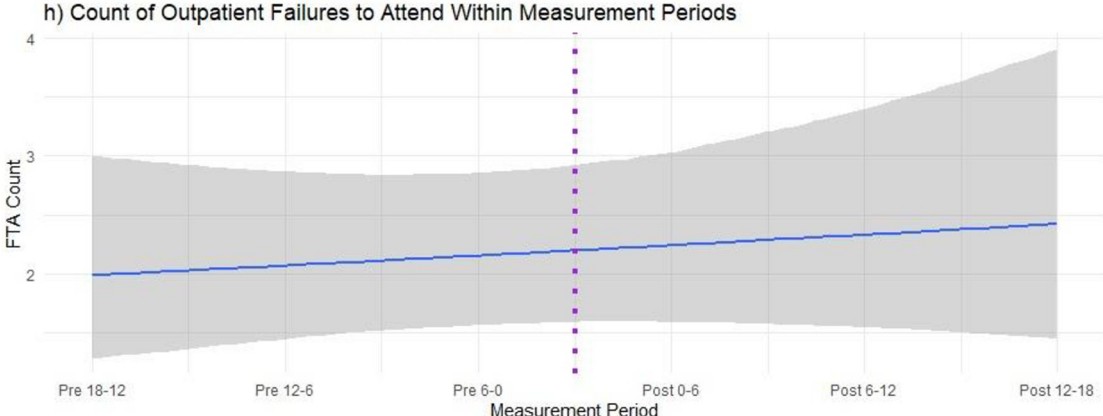

**Fig 1. Hospital statistical analysis—Failure to attend.**

In order to have greater understanding of why there was no impact on FTA in the quantitative data, the research team undertook the thematic reanalysis of the qualitative data.

Three major themes were identified: Access barriers; Failure to recognise personal stigma of FTA; and, Bridging the gap.

## Access barriers

The disparity between the patient needs and the logistics of health service provision indicated the conflicting priorities and access barriers of each of the key stakeholders. Patients and nurse navigators reported being frustrated by rigid service processes that did not recognise the unique challenges of chronically unwell people to meet the physical and financial requirements for attendance.

Common barriers to attending outpatient appointments were identified, leading to patient anxiety often due to a lack of communication between the health services and the patients. For example:

*A referral was received from [specialist] due to [patient's] complex medical needs. It was reported that [patient's wife] was struggling to coordinate multiple medical appointments in [regional town] and Brisbane. Lack of communication between [patient] and [wife], GP, [regional hospital] and [Brisbane Hospital] had been identified as an issue in providing follow up in the community*

*(nurse navigator).*

Patients expressed feelings of rejection when told that they were no longer on the waiting list for an outpatient appointment particularly if they felt they had tried within their means to attend.

*I had lots of appointments but no transport. Sometimes I had appointments almost every day. I had no transport and could not afford the transport to get there. Due to my anxiety I don't like talking on the phone. I know I should have but I didn't ring to say I wasn't coming. I know they [outpatients] got angry with me and think I don't care, but I was so sick and no money. When I did go to appointments, they treated me like I was a lost cause*

*(Patient).*

The reasons for non-attendance went beyond the individual patient. Organisational or process errors were also identified as causative factors, as described by this nurse navigator:

*On investigation it was found that the referral process was not smooth, and the referral was refaxed 3 times due to– 1) not being readable, 2) Not being received and on the 3rd time it was emailed directly to the [coordinator] for support*

*(nurse navigator).*

There is no apparent consideration of patient related impediments beyond their control, such as financial or physical factors that preclude patients from attending appointments, as one nurse navigator described:

*They [patients] have often seen lots of doctors, lots of service providers in hospital and then they leave hospital and they have no contact with the person [specialist] until they have an outpatient appointment. Sometimes the outpatient appointment is a quick turnaround, and so they don't even know that that appointment has been made for them until they've missed it and got a letter saying they failed to attend*

*(nurse navigator).*

The logistical challenges of attending multiple appointments in multiple locations, often long distances away from the patient's home were identified by the nurse navigators.

*Appointments between January and June totalled six trips to and from the Royal Brisbane (for surgery and reviews). Patient paid taxi fares to and from [specialist outpatients] (not covered by Patient Transport Scheme). Seven appointments in [town] outpatients often occurring within a couple of days following Brisbane appointments or clashing with Brisbane appointments. The emotional, physical and financial impact was very high*

*(Nurse navigator).*

Scheduling for the outpatient appointments occurs at regional health services according to availability of specialists. When specialists are not available in a regional health service, patients are required to travel to Brisbane, with distances being, for some patients, in excess of 1000km in a round trip. Nurse navigators working in regional health services frequently raised issues related to the difficulties in the cost of, and access to, travel options for these patients. For example, a nurse navigator identified the issues facing one patient regarding appointments that were geographically dislocated;

- *The financial struggle caused by the number of trips to Brisbane*;

- *Return trip from Brisbane often had [patient] returning to [regional town over 1000km from Brisbane] after dark. [Patient's wife] was not able to pick him up due to night blindness*;

- *Number of appointments booked for different specialities and health services*; *and*

- *Concerned he [patient] would not be back from Brisbane in time to attend his [regional health service] appointment and have [therefore] FTA recorded (Nurse navigator)*

Although the nurse navigators identified that they try to utilise telehealth as often as possible, they noted that some specialists will not use this virtual technology, whilst access to it in

some of the more remote locations is not possible due to the availability of bandwidths, or patients do not have access to the technology. Notwithstanding this, the standard says that:

... "health services must ensure that the best interests of the patient take precedence over the interests of the health service. This includes not staggering appointments over a number of days when scheduling clinic appointments for more than one specialty and by coordinating appointments, so they are on the same day whenever possible. Where a patient is booked for a multidisciplinary clinic appointment, health services must ensure that all care provided for the patient occurs in a single clinic appointment" [9]

(p. 20).

Nurse navigators identified a lack of communication between the various agencies involved in appointments, noting the difficulties in coordinating multiple appointments:

*A phone call I had yesterday. I have a young lady, she's got an appointment at quarter past nine one day, on that same day she's got a 3.30 pm appointment on a different team. She said look, you know, I'll come for one, but not the other*

*(Nurse navigator).*

For patients, multiple appointments challenged their attempts at normalising life, with the out-of-pocket financial burden, logistics of travel and often having to rely on family and friends to assist in transport being common issues. It left patients frustrated and demoralised as this patient noted:

*I was forgetting appointments. My son took me to appointments when he had a day off but when he had to work, I had trouble getting to appointments. Sometimes I lost the appointment letter or forgot to tell [son]. When I did go to appointments, I couldn't hear what they said. Because of the accents of a lot of the doctors and my poor hearing, I can't hear or understand what they are saying. I don't think they understood what I was asking them either*

*(Patient).*

At times, patients became overwhelmed with appointments, forgot, or had not been aware of the appointment in order to attend.

*Sometimes they will say FTA and the patient says I don't know what that means, and they'll say you haven't attended [x] amount of appointments, and most of the time they'll say I didn't know about it and there will be a legitimate reason. Sometimes there's not*

*(Nurse navigator).*

## Failure to recognise personal stigma of FTA

Routinely in healthcare settings, staff use discipline/area terminology as part of their professional vocabulary. Further, abbreviations in documents, particularly patient notes, are key to communication across services, of which FTA is one. It is seemingly insignificant for a clinician/hospital administrator to write FTA in a patient's notes. However, this seemingly insignificant note can have deep and lasting impacts on the patient. For example:

*She [patient] would come into hospital several times a month looking for reassurance and company. Most of the time she would leave without treatment. I have often contacted the different clinics to re-book her appointments as she FTAs with them. I have been meeting her at her appointments and helping her with her high anxiety and panic attacks*

*(Nurse navigator).*

Both nurse navigators and their patients verbalised the negative label attached to patients with FTAs, saying that regardless of the reason for the FTA, patients are made to feel accountable for not attending, describing the commonly referred to 'two strikes and you are out' rule as a personal penalty rather than an administrative activity that could be better managed.

*I feel like a lot of patients do put a label on FTA as well, and they are quite offended when the doctor will say, well, you've FTA'd x amount of times and they will go, actually, I didn't actually receive the letter or the text message. There seems to be a barrier there*

*(Nurse navigator).*

Sometimes the failure is just that the patient is too sick to respond. For example:

*There's no consistency with the notification of appointments, some people get letters, some people get text messages. . .I don't even want to think where he [patient] would have ended up. He was quite emaciated when we found him. But he's another example of the FTAs, and it sort of makes you think how many more out there are deemed failed to attend because they don't reply to phone calls or the letters*

*(Nurse navigator).*

Nurse navigators identified how, with minimal coordination, advocacy and time spent finding out the issues related to FTAs, solutions can be found. Additionally, nurse navigators expressed that they spend a considerable amount of time with their patients, including attending their appointments with them. For example:

*I think simple things like people failing to attend their appointments because they can't afford to get transport. So just letting them know that this is their pay week, so you can make appointments on pay week. Something simple like that, but patients are constantly marked as FTAs and they get branded like that. There's normally a solution. Some people just don't turn up, but the majority of them you can facilitate them attending*

*(Nurse navigator).*

The 'failure' label was shown to lead to further mistrust and frustration with the healthcare system, which can then have long term effects on patient engagement with the health service. The idea that the patient 'failed' discounted the person as a human being.

*And the word failed, it does, it's a negative, and these people are already in a negative place in their lives and then we go and put on another little label on them, and sometimes it's not a great result*

*(Nurse navigator).*

Nurse navigators disapproved of the term FTA and described the need to change the language within the service to be more inclusive and welcoming to their cohort of patients.

*It's something that we've had on the table for a very, very long time, but we need to–what we're doing now is not working, so who cares if we do something else that fails, because we're failing already*

*(Nurse navigator).*

## Bridging the gap

The first key role principle of the nurse navigator service is to coordinate person-centred care [8], thus a strong theme emerged within the nursing narrative about the role they play in delivering care to individuals. The nurse navigator narratives demonstrated the important role they play in bridging the gap between medical, allied health, community and acute services. They consistently spoke of how they helped a patient to change, align or reschedule appointments, describing how difficult it was for patients to coordinate multiple appointments across numerous services, geographically distant from each other. Their role is one of being an advocate for patients, bridging the gap through information sharing and discussion with service providers alongside, and on behalf of, their patients. As one nurse navigator said:

*It has taken a lot of work, liaison and effort to encourage other health care providers and services to see [patient] as a person with human problems*

*(Nurse navigator).*

Nurse navigators attempted to ensure that patient appointments not only aligned with the patient's situation, but that what was said at those appointments is understood as well. For example:

*She [nurse navigator] help with doctor appointments for [patient], going with him and speaking with him and relaying doctor speak to [patient]*

*(Patient's carer).*

*I liaised with the Senior Medical Officer and we implemented a treatment plan. I also discussed with the patient what her goals of treatment were and also her concerns, and experience with [condition] in the past. I also provided education regarding the service and what she could expect over the coming months*

*(Nurse navigator).*

Nurse navigators demonstrated a level of authority within the system that allowed them to negotiate change, reschedule appointments and provide a cohesive solution for the person. As such, they have been key to changing the dynamics of the patient/health worker relationship, by ensuring the patient is central in the care process.

*So many appointments and so many clinicians involved in her care. I commented to her that I can see that for many years a lot of people have been telling her what she should be doing and*

*how she should be doing it but I wondered if anyone had actually stopped and asked [patient] what she thinks or what she wants. She smiled at me and said, 'you're right'*

*(Nurse navigator).*

The lack of consideration for a person's situation regarding attending outpatient appointments and the advocacy that nurse navigators provide, was described as:

*A lot of departments don't talk to each other. . .this morning I had a fellow that was admitted over the weekend and transferred over to [hospital] and he had [multiple outpatient appointments] so I have . . .got it cancelled for today and next week, and had them rescheduled. If I hadn't have done that, he would have been put down as a failed to attend, and that would have gone on his record. So, no one would have really checked, and I guess just even basic things like reading people's charts and finding out that information, there's no follow-up. And departments, like I said, just don't talk to each other*

*(Nurse navigator).*

The impact that nurse navigators have on supporting patients to gain confidence in themselves to deal with issues such as dealing with appointments is highlighted in this comment:

*I look forward to going to appointments because I can see the good side of it now and can ask questions and understand what they are saying*

*(Patient)*

The reality of this advocacy and coordination is that it supports effective coordinated care, linking patients to appointments that are needed and, in a time and place that is best for the patient:

*Coordination of appointments to a more appropriate time of day to allow patient a chance to attend. Nurse Navigator liaised with ENT complex care coordinator in Brisbane hospital about upcoming appointment. Physiotherapy post neck surgery overlooked–NN contacted RBWH to refer patient to physiotherapy post-surgery–NN support provided during appointments until comfortable in attending independently*

*(Nurse navigator).*

## Discussion

This paper explored the phrase 'Failure to Attend' applied to outpatient appointments, from the perspective of nurse navigators and their patients. The paper reports on quantitative data which showed no improvement and then used qualitative data to explore this. Our analysis highlighted a disconnect between what is recommended as standards of practice in managing outpatient appointments and what is happening with patients attending (or not attending) outpatient appointments, within Queensland Health. This research has demonstrated that the language used by clinicians is 'Failure to Attend'. Within the hospital documentation, the term 'No show' is also used. Whether missed appointments are labelled as 'no show' or FTA, the connotations are negative and there is a clear disconnect between the standard and the practice of how FTAs are managed. The patients see themselves as the victim in a system that they cannot control or align with, in their attempts at normalising life amidst multiple appointments

across a myriad of services and geographic locations. For them, the system has failed. In contrast, for the health system, the patient has failed to take up the support and help offered. This research demonstrates that appointments are a top priority for the health service but may be a low priority for people with chronic conditions for various reasons, none of which are intentionally obstructive to the effective functioning of health services and efficient use of finite resources.

Scheduling of appointments is done to fit the routine of the specific clinic/health professional rather than adopting a whole of service approach or person-centred approach, as the outpatient standards intend. Individual patient priorities influence decision-making about what appointments they will attend, and in many instances, what patients prioritise would not be taken into consideration when service providers are scheduling appointments. Several patients reported that lack of communication or the lack of ensuring that information was relayed to them in a timely manner had a significant impact on outcomes. These findings support those of Roberts and Callanan [3] who found that 12% of patients attending outpatient appointments are FTAs for all the reasons provided by our study cohort, while a further 10% of FTAs are the result of systemic issues. When examining the pivotal coordination role that nurse navigators provide, we argue that, with few exceptions, the FTA is merely one symptom in the miscommunication that occurs between health services and patients with complex care needs.

For patients, the anxiety that is created as a result of FTA, for whatever reason, was evident. The word 'failure' conjures up issues of being a 'bad' patient as verbalised by the patients in our study. Yet health services used the term 'failure' to describe the actions (or non-actions) of patients routinely. For example, 'failure to attend', 'failure to progress', 'failure to cope' or in the maternity services genre, 'failure to thrive/breastfeed' and 'failed trial of labour'. Often, these terms are intermingled with the concepts of compliance and non-compliance in care [18]. Furber and Thomson [18] found that 'failure' language used in birthing women had a negative effect on the patient/clinician partnership. Additionally, stigmatised language written in records influences clinician behaviour and decision-making about how to communicate with the mother and what treatment to provide [19]. Goddu et al [19] highlighted that where stigmatised language was used in patient records, the patients were subjected to less vigorous pain management. Conversely, "good" communication and language in all mediums can 'lower anxiety, build confidence, educate and help to improve self-care' [20], (p.1630).

The language that health professionals use in communication with patients matters [21]. Language constructs the world around us and perpetuates the natural social order within any given institution [22]. Language can inspire and motivate or alienate and demean [23]. A good example of this is the changing nature of the term "compliance", which has evolved to "adherence" and more recently "concordance". Models of care that advocate for concordance are more person-centred [23]. The language used has the power to impact upon patient care, and in the case of the nurse navigators, those patients who are most vulnerable within the health-care system.

When offering care to patients, health services are in a constant state of tension between the rights of the individual and the fiscal management of those services. The debate around sustainable universal health care services centres on economic, social and political sustainability [24]. Public health services in Australia offer universal health care and thus have an obligation to offer access to services in a range of formats whilst also ensuring the management of these services is the responsibilities of the States and Territories. Increasingly health services are changing the way they include patients in decision-making, making standards of practice and policies more interactive and person-centred [9, 25]. While the patient is on the public health service outpatient waiting list, the health service has a duty of care to ensure that "reasonable

efforts to provide specialist outpatient care within clinically recommended timeframes, communicating with patients, referring practitioners and nominated general practitioners, and responding to information regarding changes to the patient's condition during this time appropriately" [9] (p. 8). According to the standards, the patient also has responsibility for their own care. For example, a patient is responsible for communicating their inability to attend an appointment with the clinic within 14 days and to re-book their appointment. If the patient does not do this, the service is directed to review the referral and potentially remove the patient from the clinic list [9]. How this is communicated is less clear, however, health services must ensure equitable access to those on any waiting lists, and thus the management of FTA does become necessary.

The reality of life for people with multiple chronic conditions is that attending appointments and specialists can be a very difficult task. Roberts and Callanan [3] found that reasons for not attending appointments were multifaceted and included forgetting the appointment, fear of rejection, clerical booking errors and not being aware of the appointment. A patient who does not attend an outpatient appointment does not always do so willingly. Unfortunately, the process for managing services and FTAs is such that when a patient is unable to attend, they can lose their place in the queue, needing to start the referral process again through their GP. While it certainly may be the case in some situations, not all patients walk away from this responsibility willingly, and the choice to be removed from the outpatient list is not always theirs to make. An FTA in the outpatient setting is a black mark, in that missing two appointments provides the avenue for discharge from the service: two strikes and you're out [9]. In one of our case studies the letter informing patients of the patient's discharge was a faceless and nameless process. The letter was not signed by any clinician or individual and was simply signed from the 'outpatient department.' This suggests that while the decision to remove the patient has been firmly made, the is no one assigned responsibility for this process, other than the patient. This represents a closed door to the patient.

The process of recording an FTA in the system, initiating a clinical review and communicating the outcome of the FTA to the patient appears to be disjointed, clerically driven and not employed for its intended purpose [9]. Outpatient appointment administrative staff members are tasked with recording the patient's status (checked in, cancelled, client seen, FTA, did not wait). Where some digital systems allow for a reason to be recorded in a free text column of the system, the judgement of this is at times left to a non-clinical staff member. It is unclear how this information is obtained and interpreted and in what way it is reviewed to improve the service. It is also unclear as to how appropriate this duty is for clerical staff, given the impact that this label can have on patients who are already socially and clinically compromised. Additionally, while the specialist outpatient standard states that the reason for an FTA should be recorded, it is not clear how hospital administrators use these data within a quality improvement framework to improve patient outcomes. The process of managing outpatient services should have a focus on appropriate use of clinical services, on clinical need and on reducing barriers to access, and data around the reasons for an FTA could provide valuable insights to this.

The question that is posed from our research is: would the patient have 'failed' in the context of a collaborative and person-centred system? As providers of health care, clinicians have an obligation to partner with individuals and to ensure processes meet the needs of the person [26]. Ultimately, responsibility and blame do little to ensure that the person receives the appropriate care, and that services are used in the most efficient way. This is particularly important for rural and remote area located people living outside of major treatment hubs. The cost of travel is well documented. As noted by the Australian Institute of Health and Welfare [27], "in 2016, people in remote areas were more likely to report barriers accessing GPs and specialists

than [people living in] major cities". They also noted that some patients may relocate to a less remote area to facilitate effective treatment, though this was not explored in our study.

Health service staff need to move past who is right or wrong and redirect this to a conversation about how to engage people with services, and put the patient's opinions, feelings and thoughts at the centre of care planning. Nurse navigators are bridging this communication gap through their advocacy role in communicating the patient's needs and stepping in to resolve issues, such as the FTAs that are not directly the patient's 'fault'. This research advocates for a change to the term 'Appointment did not proceed'. This may have more positive connotations for the patient and may direct the conversation from blame, to asking why the appointment was not able proceed. Additionally, investigation into the processes of obtaining, recording, interpreting, and evaluating this data may be of considerable benefit to health services and to patients attending the service.

## Limitations

The paper does not explore the views of clinicians within the various health services or the difficulties they experience in coordinating services. This research is limited to data collected in one Australian state and to the public health provider sphere. This research has reviewed data associated with those with multimorbidity only as the cohort within the nurse navigator service who have multiple chronic conditions and hence more frequent touch points with outpatient services.

## Conclusion

The label of FTA poses a risk to further disengage people from service utilisation. This paper raises the awareness of the language used inadvertently by health service providers, whereby patients are apportioned blame when they 'fail' to comply with health service directives. We examined the concept of 'failure to attend' outpatient specialist appointments and identified extraneous issues that prevented patients from attending an outpatient appointment. Once a patient received a label of FTA, they were removed from the specialist outpatient waiting list if this occurred more than twice, regardless of the reason. People with multimorbidity have complex medical and social needs, that need to prioritise daily the resources they have. Nurse navigators work within a person-centred model to ensure that the person is fully engaged with the health decisions made and then assist them in aligning their life with the multiple specialist appointments. As such, nurse navigators are a central point for communication and liaison between patients and the health system.

## Implications for health policy and practice

This article advocates for the consideration of alternative language, namely 'Appointment did not proceed', to be used when referring to a patient's inability to attend an appointment, and to develop collaborative methods of communication, management of outpatient appointments and engagement with multimorbid patients. This may have relevance for patients who live in rural and remote areas who continue to experience financial and physical disadvantages compared with people living in major centres.

## Supporting information

**S1 Checklist. COREQ checklist.**
(DOCX)

## Author Contributions

**Conceptualization:** Amy-Louise Byrne, Adele Baldwin, Clare Harvey, Janie Brown, Desley Hegney, Bridget Ferguson, Brody Heritage, Shona Thompson, Janine Palmer.

**Data curation:** Amy-Louise Byrne, David Heard.

**Formal analysis:** Amy-Louise Byrne, Adele Baldwin, Clare Harvey, Eileen Willis, Bridget Ferguson, Doug Kynaston.

**Funding acquisition:** Clare Harvey.

**Methodology:** Amy-Louise Byrne, Adele Baldwin, Clare Harvey.

**Validation:** Janie Brown, Eileen Willis, Desley Hegney, Bridget Ferguson, Jenni Judd, Rachel Forrest, Brody Heritage.

**Writing – original draft:** Amy-Louise Byrne.

**Writing – review & editing:** Amy-Louise Byrne, Adele Baldwin, Clare Harvey, Janie Brown, Eileen Willis, Desley Hegney, Bridget Ferguson, Jenni Judd, Doug Kynaston, Rachel Forrest, Brody Heritage, David Heard, Sandy Mclellan, Shona Thompson, Janine Palmer.

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
