## [Decision Letter · Decision Letter 0]

21 Aug 2020

PONE-D-20-11849

Nurse navigators and their patients: Exploring the phenomenon of ‘Failure to Attend’ and its effects on continuity of care

PLOS ONE

Dear Dr. Byrne,

Thank you for submitting your manuscript to PLOS ONE. After careful consideration, we feel that it has merit but does not fully meet PLOS ONE’s publication criteria as it currently stands. Therefore, we invite you to submit a revised version of the manuscript that addresses the points raised during the review process.

We look forward to receiving your revised manuscript.

Kind regards,

Janhavi Ajit Vaingankar

Academic Editor

PLOS ONE

Journal Requirements:

2.Thank you for including your ethics statement:  "A multiple site ethical approval was awarded to the project in March 2018

HREC/18/QTDD/8".   

3.Thank you for stating the following in the Funding Section of your manuscript:

[This project has been funded by Queensland Health, Australia.]

 [The funders had no role in study design, data collection and analysis, decision to publish, or preparation of the manuscript.]

Reviewers' comments:

Reviewer's Responses to Questions

**Comments to the Author**

1. Is the manuscript technically sound, and do the data support the conclusions?

Reviewer #1: Partly

Reviewer #2: Yes

2. Has the statistical analysis been performed appropriately and rigorously? 

Reviewer #1: N/A

Reviewer #2: Yes

3. Have the authors made all data underlying the findings in their manuscript fully available?

Reviewer #1: Yes

Reviewer #2: Yes

4. Is the manuscript presented in an intelligible fashion and written in standard English?

Reviewer #1: Yes

Reviewer #2: No

5. Review Comments to the Author

Reviewer #1: The aim of the presented study is to capture the reasons and define the concept of “Failure to Attend” from the perspectives of patients and nurse navigators. Nurse navigators ensure continuity of care of individual patients with multiple and complex chronical conditions by improving case and care management and this include facilitating self-management and supporting compliance of the patients, with the aim to reduce preventable hospital admissions and improve patient outcomes. With regard to this, FTA is defined as an indicator of hospital utilization. This study is part of a broader interim evaluation of the nurse navigator service and therefore the authors secondary analyse interview data with the thematic focus on FTA.

The authors explore the phenomenon “Failure to Attend” which is very relevant concerning to patients’ access to and continuity of care and the study also make a contribution to the actual discussion of inappropriate language used in care setting (e.g. discussion of “compliance”). But they implicitly also describe another phenomenon: The difference between Talk and Action in Health Care Settings which helps to reduce complexity.

Despite that, some shortcomings exist and the paper has to be strengthen before publication to improve quality. Therefore, some suggestions and recommendations are following:

- Introduction/Background -

The focus is more on the project setting and context but I recommend to take also the evidence of the used theoretical concepts into account (e.g. Continuity of Care and its associations to nursing case management or nursing led transition management; e.g. state of the research onto FTA)

- Aim -

The authors provide divergent phrasing of the aim:

“The aim of this paper is to capture the reasons for ‘Failure to Attend’ (FTA) through the narrative of nurse navigators and their patients and to explore how it is defined by the patient.” (32/33) (thematic coding/content analysis is appropriate)

“The aim of this paper is to provide an understanding of the concept, use and impact of the phrase ‘Failure to Attend’ on the relationship between health services and navigated patients.” (140/141) (a deeper, more interpretative method would be appropriate)

This is not only a divergence in wording but rather in the appropriateness of the methods used for data analysis (please see the hints within the brackets).

n my view, there is no good fit between the different aims, the research questions, the methods used and the reported results. Thus I recommend to adapt the divergent aims and rephrase it consistently, in reference to the methods used. In that regard I also reccommend to customize the research questions whilst taking the results into account.

- Methodological issues -

It is not clear how each of the 15 authors contributed to the study or this paper. Regarding this, the COREQ-checklist is also fulfilled in an insufficient way and the given information did not create transparency. (e.g. Who conducted the group and individual interviews? Data were collected “…by members of the research team”. Is the research team of this study identical as the team who conducted the evaluation?). I recommend to clarify the contribution of the authors and take the COREQ guideline into account in reporting this qualitative research.

The reported research design describes the interim evaluation of the nurse navigators approach. Interviews were conducted as part of this evaluation and the transcribed interviews were analysed with the thematic focus on FTA (“the management of FTAs and the effects this label has on patients when they are unable to attend specialist outpatient appointments.” (170/171)). It is unclear if FTA was explicitly included into the data collection (e.g. as theme within the interview guide) in an iterative way or if primarily collected data of the nurse navigator evaluation were analysed with the new lens of FTA? I recommend to clear the difference explicitly if it was an explanatory sequential mixed-method design including FTA as theme primarily OR a secondary analyses of the qualitative evaluation data? I think it was the second one, but in my view the manuscript is ambigous with regard to that.

Data analyses: A combined deductive inductive approach was used to analyse data and out of this procedure a code system evolved. I recommend to report the code system (e.g. as mind-map, code tree or table) to ensure transparency and to guide the reader through the results by visualization.

Description of data analysis is a bit confusing: “Second cycle coding was then completed by the research team members collaboratively, and codes were … further developed within the analytic lens of the nurse navigator service” (208-210). Why did the authors not use the lens of FTA?

- Ethics -

I recommend to explicitly mention the research ethics committee, not only the reference number.

- Results -

Conflicting patients’/health services priorities.

In my view, the authors report predominantly reasons of failure and they identified access barriers (e.g. money, time, distance, disability, equipping). I cannot comprehend the label of this major code “Conflicting patients’/health services priorities”. I recommend to provide a definition or meaning of the major code labels to clarify this.

Failure to Recognise Personal Stigma of FTA.

Only narratives of nurse navigators were quoted. Is this in line with the aims and research questions of the study?

Bridging the gap.

Nurse navigator’s perspectives were interpreted as patient centred by nursing researcher and based on this interpretation this was reasoned as bridging behaviour. I recommend to reconsider this section and keep the difference in talk vs. action (Brunson / Olson 1993) into account. I also recommend to frame this result explicitly as nurse perspective.

- Discussion/Conclusion -

The authors offer a successful discussion and adequate conclusion. They discuss “Failure to attend” in contrast with other concepts, e.g. "compliance", and they argue to differ the phrase “Failure to attend” from the phrase “Appointment did not proceed” and the last one should be used when referring to a patient’s inability to attend an appointment

Reviewer #2: Manuscript purpose to capture reasons for "failure to attend" is not truly satisfied through your research questions - How is the phrase FTA described? Priorities of providers that impact patients capacity to attend? and Impact of the FTA label? These are important research questions and should be reflected in the manuscript purpose.

Very timely topic and important area of inclusion to the literature - patient centeredness is important in all aspects as you note.

A few minor grammatical issues - example line 565

6. PLOS authors have the option to publish the peer review history of their article (what does this mean?). If published, this will include your full peer review and any attached files.

Reviewer #1: **Yes: **Stefan Nöst

Reviewer #2: **Yes: **Connie Cole DNP, RN-BC, NP-C

---

## [Author Response · Author response to Decision Letter 0]

13 Sep 2020

Please see attached file for full revisions

I have now amended the research design heading to read; 'Research design and Methods' and the ethics statement sits within this

---

## [Decision Letter · Decision Letter 1]

8 Dec 2020

PONE-D-20-11849R1

Understanding the impact and causes of 'failure to attend' on continuity of care for patients with chronic conditions.

PLOS ONE

Dear Dr. Byrne,

Thank you for submitting your manuscript to PLOS ONE. After careful consideration, we feel that it has merit but does not fully meet PLOS ONE’s publication criteria as it currently stands. Therefore, we invite you to submit a revised version of the manuscript that addresses the points raised during the review process.

We look forward to receiving your revised manuscript.

Kind regards,

Janhavi Ajit Vaingankar

Academic Editor

PLOS ONE

Reviewers' comments:

Reviewer's Responses to Questions

**Comments to the Author**

1. If the authors have adequately addressed your comments raised in a previous round of review and you feel that this manuscript is now acceptable for publication, you may indicate that here to bypass the “Comments to the Author” section, enter your conflict of interest statement in the “Confidential to Editor” section, and submit your "Accept" recommendation.

Reviewer #1: (No Response)

2. Is the manuscript technically sound, and do the data support the conclusions?

Reviewer #1: Partly

3. Has the statistical analysis been performed appropriately and rigorously? 

Reviewer #1: N/A

4. Have the authors made all data underlying the findings in their manuscript fully available?

Reviewer #1: Yes

5. Is the manuscript presented in an intelligible fashion and written in standard English?

Reviewer #1: Yes

6. Review Comments to the Author

Reviewer #1: Thank you for reviewing this manuscript a second time. The authors processed my suggestions for the most part which now led to an improved and sharpened manuscript, but some minor shortcomings remained:

From a methodological view Ich recommend to specify the sample size (counts of conducted nurse navigator interviews as well as patient interviews) of the primary qualitative study which were now secondary analyzed.

With regard to this, I also suggest to describe the sample chareteristics at a glance, within the result section.

Different terminology was used to cite the quotes: "Nurse Navigator", "nurse navigator" and "Navigator"). I recommend to harmonize this and with regard to transperancy and in order to demonstrate variety within the sampling, I also suggest to use pseudonyms and not only roles to cite the quotes .

As already mentioned in the first review I suggest to vizualise the categorization/code system with the aim to maintain and guide the reader.

7. PLOS authors have the option to publish the peer review history of their article (what does this mean?). If published, this will include your full peer review and any attached files.

Reviewer #1: **Yes: **Stefan Nöst

---

## [Decision Letter · Decision Letter 2]

19 Jan 2021

PONE-D-20-11849R2

Understanding the impact and causes of 'failure to attend' on continuity of care for patients with chronic conditions.

PLOS ONE

Dear Dr. Byrne,

Thank you for submitting your manuscript to PLOS ONE. After careful consideration, we feel that it has merit but does not fully meet PLOS ONE’s publication criteria as it currently stands. Therefore, we invite you to submit a revised version of the manuscript that addresses the points raised during the review process.

We look forward to receiving your revised manuscript.

Kind regards,

Janhavi Ajit Vaingankar

Academic Editor

PLOS ONE

Reviewers' comments:

Reviewer's Responses to Questions

**Comments to the Author**

1. If the authors have adequately addressed your comments raised in a previous round of review and you feel that this manuscript is now acceptable for publication, you may indicate that here to bypass the “Comments to the Author” section, enter your conflict of interest statement in the “Confidential to Editor” section, and submit your "Accept" recommendation.

Reviewer #1: (No Response)

2. Is the manuscript technically sound, and do the data support the conclusions?

Reviewer #1: Partly

3. Has the statistical analysis been performed appropriately and rigorously? 

Reviewer #1: N/A

4. Have the authors made all data underlying the findings in their manuscript fully available?

Reviewer #1: Yes

5. Is the manuscript presented in an intelligible fashion and written in standard English?

Reviewer #1: Yes

6. Review Comments to the Author

Reviewer #1: Thank you for your revision which enhanced the quality of the manuscript. But I would repeat some minor suggestions to improve research transparency of your qualitative research report:

1. Regarding to my first review: the COREQ-checklist is further

fulfilled in an insufficient way: First, I cannot identify your responded information within the masucript for the COREQ questions 2, 3, 5. Second, your respoense to question 8 is not suitable. Third, question 8 refer to recording (possible response: "interviews are recorded and transcribed by a professional transcription service").

2. Regarding to my first and second review: the vizualisation of the results (codes and/or themes) would be very heplful for reader guidance.

3. Regarding to my second review: to demonstrate variety of the reported quotes, I suggest to use pseudonymized references (e.g. Nurse Navigator 1 vs. Nurse Navigators 8) to ensure transparency.

4. In addition to my second review: To characterize the sample usage of descriptive statistics are usual to ensure research transparency.

7. PLOS authors have the option to publish the peer review history of their article (what does this mean?). If published, this will include your full peer review and any attached files.

Reviewer #1: **Yes: **Stefan Nöst

---

## [Author Response · Author response to Decision Letter 2]

7 Feb 2021

Please see rebuttal in cover letter attached

---

## [Editor Report · Decision Letter 3]

17 Feb 2021

Understanding the impact and causes of 'failure to attend' on continuity of care for patients with chronic conditions.

PONE-D-20-11849R3

Dear Dr. Byrne,

We’re pleased to inform you that your manuscript has been judged scientifically suitable for publication and will be formally accepted for publication once it meets all outstanding technical requirements.

Kind regards,

Janhavi Ajit Vaingankar

Academic Editor

PLOS ONE
---

## [Editor Report · Acceptance letter]

19 Feb 2021

PONE-D-20-11849R3 

Understanding the impact and causes of ‘failure to attend’ on continuity of care for patients with chronic conditions 

Dear Dr. Byrne:

I'm pleased to inform you that your manuscript has been deemed suitable for publication in PLOS ONE. Congratulations! Your manuscript is now with our production department. 

Kind regards, 

on behalf of

Ms Janhavi Ajit Vaingankar 

Academic Editor

PLOS ONE